# Investigation into Deep Brain Stimulation Lead Designs: A Patient-Specific Simulation Study

**DOI:** 10.3390/brainsci6030039

**Published:** 2016-09-07

**Authors:** Fabiola Alonso, Malcolm A. Latorre, Nathanael Göransson, Peter Zsigmond, Karin Wårdell

**Affiliations:** 1Department of Biomedical Engineering, Linköping University, Linköping 58185, Sweden; malcolm.latorre@liu.se (M.A.L.); Nathanael.Goransson@regionostergotland.se (N.G.); karin.wardell@liu.se (K.W.); 2Department of Neurosurgery, Linköping University Hospital, Region Östergötland, Linköping 58185, Sweden; Peter.Zsigmond@regionostergotland.se; 3Department of Clinical and Experimental Medicine, Linköping University, Linköping 58185, Sweden

**Keywords:** deep brain stimulation (DBS), steering, patient-specific, electric field, finite element method, neuron model, brain model, zona incerta (ZI), electrode design

## Abstract

New deep brain stimulation (DBS) electrode designs offer operation in voltage and current mode and capability to steer the electric field (EF). The aim of the study was to compare the EF distributions of four DBS leads at equivalent amplitudes (3 V and 3.4 mA). Finite element method (FEM) simulations (*n* = 38) around cylindrical contacts (leads 3389, 6148) or equivalent contact configurations (leads 6180, SureStim1) were performed using homogeneous and patient-specific (heterogeneous) brain tissue models. Steering effects of 6180 and SureStim1 were compared with symmetric stimulation fields. To make relative comparisons between simulations, an EF isolevel of 0.2 V/mm was chosen based on neuron model simulations (*n* = 832) applied before EF visualization and comparisons. The simulations show that the EF distribution is largely influenced by the heterogeneity of the tissue, and the operating mode. Equivalent contact configurations result in similar EF distributions. In steering configurations, larger EF volumes were achieved in current mode using equivalent amplitudes. The methodology was demonstrated in a patient-specific simulation around the zona incerta and a “virtual” ventral intermediate nucleus target. In conclusion, lead design differences are enhanced when using patient-specific tissue models and current stimulation mode.

## 1. Introduction

Deep brain stimulation (DBS) is an established technique to alleviate the symptoms caused by several movement disorders such as Parkinson’s disease and essential tremor. DBS is now also expanding towards other symptoms such as psychiatric illness [1]. The technique has been proven to be successful even though the mechanisms of action are still uncertain, which makes it difficult to have complete control on the desired effect and avoid side effects.

Traditionally, DBS systems have operated in voltage mode using conventional ring-shaped electrodes generating a symmetrical stimulation field around the lead. Recently, new electrode designs offer the capability to steer the stimulation field allowing some compensation for a possible lead misplacement [2,3]. The operating mode has also been modified delivering current instead of voltage stimulation. Current controlled systems, in comparison to voltage, automatically adjust the voltage to changes in the surrounding tissue impedance, in order to deliver a constant current [4]. Brain tissue is an electrically conductive medium in which the distribution of the electric field (EF) can be calculated and visualized with computer models that solve the corresponding differential equation. In this study, the finite element method (FEM) has been used to evaluate and compare the EF from four different leads used in DBS systems.

Numerous computational models have been used to predict and visualize the first derivative of the electric potential, i.e., the EF [5,6,7,8] or the second derivative of the electric potential generated by DBS systems [9,10]. However, these are usually employed using traditional leads with voltage control operating mode. We have previously compared the conventional leads *Medtronic 3389* and *St. Jude Medical 6148* in different operating modes and time points [11]. Other simulations studied the influence from heterogeneity and anisotropy for the 3389 lead [12,13]. This study extends the comparisons to include two steering field leads, *St. Jude Medical 6180* and *Medtronic SureStim1*. When comparing FEM simulations, a fixed isolevel EF has been useful in making relative simulation comparisons for the 3389 lead [5,6,8]. In a previous study [14], neuron model simulations were run for a range of stimulation amplitudes, pulse lengths and axon diameters. These settings and physiological parameters should be taken into account in the choice of isolevel.

The aim of the study was to compare four DBS lead EF distributions in both voltage and current modes as presented in homogenous and heterogeneous, i.e., patient-specific, tissue models for the zona incerta (ZI) and the ventral intermediate nucleus (VIM) brain targets. Furthermore, steering effect simulations were investigated and compared with conventional 3389 lead EF. Visualization of the 3389 EF for the implanted ZI target with the patient-specific stimulation settings was demonstrated.

## 2. Materials and Methods

### 2.1. Patient Data, Surgery and Imaging

DBS data and images from one patient with tremor dominant Parkinson’s disease implanted in the ZI at the Department of Neurosurgery, Linköping University Hospital were included in the study. An additional “virtual target”, VIM, along the planned trajectory was used for the simulations. Informed written consent was received from the patient and the study was approved by the local ethics committee in Linköping (2012/434-31).

Prior to surgery and under general anaesthesia, the Leksell Stereotactic System (G frame, Elekta Instrument AB, Linköping, Sweden) was attached. Thereafter, a 3 Tesla, T2-weigthed magnetic resonance imaging (MRI) Philips Intera, Eindhoven, The Netherlands) with 2 mm contiguous axial slices (2 × 0.5 × 0.5) mm^3^ was performed. Direct anatomical targeting was planned using Surgiplan^®^ (Elekta Instrument AB). Surgery followed the routine protocol [15] for DBS implantation and was completed in a single procedure. The probe’s position was verified by intraoperative fluoroscopy (Philips BV Pulsera, Philips Medical Systems, Eindhoven, The Netherlands). A postoperative computer tomography (CT) was performed to confirm the lead’s positioning the day after surgery, and a second CT was taken after 4.5 weeks (chronic time point). These CT images were separately co-registered with the preoperative MRI using Surgiplan^®^. From the postoperative image artefacts, the surgeon noted the Leksell^®^ coordinates (x, y, z) of a point at the lowest contact and another reference point 10 mm above the AC-PC line along the lead axis. These coordinates were used to place the lead within the brain model. The electrode position at the chronic time point was considered for the simulations in this study. The Leksell^®^ coordinates for ZI and VIM targets were also identified for simulations.

### 2.2. FEM Modelling and Simulation

The leads and brain tissue were modelled in the FEM software COMSOL Multiphysics 5.2 (Comsol AB, Stockholm, Sweden).

#### 2.2.1. DBS Leads

The lead geometry was based on the specifications from the corresponding manufacturing companies (Figure 1). Lead 3389 (Medtronic Inc., Minneapolis, MN, USA) and lead 6148 (St. Jude Medical Inc., Saint Paul, MN, USA) consist of four cylindrical platinum iridium alloy electrodes or contacts separated by 0.5 mm of insulating material. The contacts are 1.5 mm long except for lead 6148’s distal contact which is 3 mm long and covers the tip of the lead. The lead 3389 has a diameter of 1.27 mm and a contact surface of 6 mm^2^ while lead 6148 is 1.4 mm, with a contact surface area of 6.6 mm^2^. The steering lead 6180 (St. Jude Medical Inc., Saint Paul, MN, USA) has the same dimensions as lead 3389 and similar disposition of the contacts except for the two middle contacts which are partitioned axially into three sections; a single segment of the split-ring contact has a surface area of 1.8 mm^2^. SureStim1 lead (Medtronic Eindhoven Design Centre BV, Eindhoven, The Netherlands) also has a diameter of 1.27 mm and consists of 40 elliptical contacts of 0.66 × 0.74 mm^2^ arranged on 10 rows of four contacts each, along the lead; each contact surface area is 0.39 mm^2^ [2]. The stereotactic coordinates obtained from Surgiplan^®^ from the co-registered postoperative CT along with the fiducial points of the preoperative MRI were used to calculate the Cartesian coordinates and the angle of the lead for the FEM mode. The first contact of the lead 3389 was placed at the lower point noted by the surgeon; lead 6148 and the steering leads’ locations were adjusted to match the middle point of the active contacts.

#### 2.2.2. Brain Tissue Model

Patient-specific brain tissue models were based on preoperative MRI. An in-house developed program (ELMA) [16,17] was used to convert the medical images into COMSOL FEM software readable files. With the ELMA tool, the preoperative image was cropped to a region of interest (Figure 2a), including the VIM and the ZI. Within that region, the tissue was classified into grey matter, white matter, blood or cerebrospinal fluid based on the image intensity values. Average intensity values were calculated from three slices of the preoperative image set. Finally, the electrical conductivity, σ, was assigned according to grey matter (σ = 0.123 S/m), white matter (σ = 0.075 S/m), blood (σ = 0.7 S/m) and cerebrospinal fluid (σ = 2.0 S/m). The corresponding electric conductivities for each tissue type were obtained from tabulated values [18,19] weighted with the spectral distribution of the pulse shape [20]. The conductivity for each voxel was calculated by an interpolation function which takes into account the effects of partial volumes, thus voxels with intensity levels between grey and white matter receive an electrical conductivity between grey and white matter. The result was a cuboid of about 100 mm per side (Figure 2b) containing the electrical conductivity values for each classified voxel of the preoperative MR image. The model included a peri-electrode space (PES) of 0.25 mm to mimic the electrode–tissue interface at the chronic stage [21]. The electrical conductivity assigned to the PES corresponded to the white matter assuming its similarity to fibrous tissue (σ = 0.075 S/m) which is believed to wrap around the lead at the chronic stage [22].

The electric field was calculated by the equation for steady currents:
(1)$$\nabla \cdot J=-\nabla \cdot \left(\sigma \nabla V\right)=0\;\left(A/m^{3}\right)$$
where ***J*** is the current density (A/m^2^), *V* is the electric potential (*V*). For patient-specific models, σ corresponds to the interpolation matrix extracted by ELMA. For the homogeneous model, a single σ value corresponding to grey matter conductivity was considered for the whole brain tissue. The electrodes were set in a monopolar configuration where the active contact is considered as a voltage or current source and the outer boundaries are grounded (*V* = 0 V). For the conventional leads, the third contact (C2 and C3, for Medtronic 3389 and St. Jude 6148 respectively) was active. For SureStim1 eight consecutive electrodes corresponding to ring 6 and 7 were selected, and for the St. Jude 6180 lead the contacts 5, 6, 7 constituting the third ring were active. The active contacts of each lead were driven with either 3 V or 3.4 mA which is the equivalent current amplitude for Medtronic 3389 lead in a homogeneous model (σ = 0.123 S/m). The equivalent stimulation current value was considered as that required to achieve the same electric field to the one obtained with voltage control [11]. The inactive contacts were set to floating potential (∫−n·σ∇VdS=0 (A); n×(−∇V)=0 (V/m)) and the non-conductive surfaces of the lead were set to electric insulation (n·∇V=0 (V/m)) where ***n*** is the surface normal vector. The mesh applied was physics-controlled with a denser distribution around the leads. The mesh was set to the finest resolution available resulting in more than 2,000,000 tetrahedral elements (minimum element size of 0.026 mm). For the steering configuration, a single contact (C5) was selected for lead 6180 while for lead SureStim1, four contacts in a diamond configuration (two adjacent contacts from ring 6 and one contact from ring 5 and 7 anteriorly oriented) were active. The 3D models with ~3 million degrees of freedom were solved using the iterative COMSOL built-in conjugate gradients solver.

### 2.3. Neuron Model Simulations

An axon cable model was used in combination with the FEM model. A complete description of the neuron model is found in Åström et al. 2015 [14]. FEM modelling was completed for each lead design (*n* = 16) with a stimulation amplitude of 1 V or 1 mA for both homogenous and patient-specific brain tissue models for the VIM target. The electric potential was evaluated at the axial plane around the lead’s third contact (Figure 3a). The potential lines were extracted from the medial, lateral, posterior and anterior locations from the axial plane. The potential along the 62 parallel lines separated by 0.1 mm was exported and used as input data to the cable model to calculate the neuron activation distances. Simulations (*n* = 832) were performed for a fixed pulse width (60 µs) with variation in amplitudes (0.5–5 V in steps of 0.5 V; and 0.5–5 mA in steps of 0.5 mA) and variation in axon diameters (1.5–7.5 µm in steps of 0.5 µm) (Figure 3b).

### 2.4. Electric Field Simulations

FEM simulations of the electric field (*n* = 38) were performed in different stages setting to 3 V or 3.4 mA the third contact or equivalent as previously described. First, homogenous and patient-specific tissue models were investigated solely with lead 3389 (*n* = 6). Patient-specific simulations included two targets, the ZI and the VIM. Secondly, patient-specific models (one for each target, moving the leads accordingly, approximately 4 mm along the trajectory) were used to compare the electric field achieved by the four leads (*n* = 16) for the two operating modes. The patient-specific model of the actual implantation site in ZI was also used to investigate the EF achieved by lead 3389 with the actual stimulation 1.6 V, set four and a half weeks after implantation, which relieved the patient’s symptoms. Simulations were also performed for the corresponding equivalent value in current mode (*n* = 4). At last, simulations with steering configurations for lead 6180 and SureStim1 were performed (*n* = 8). For investigation of the steering function, additional simulations (*n* = 4) were performed for St. Jude 6180 and SureStim1 and compared with the Medtronic 3389 lead.

### 2.5. Data Analysis

The neuron model simulation output is a table of activation distances (mm) which can be presented as plots against the stimulation amplitudes (Figure 3c,d). The average deviation in activation distances between the leads was calculated as mean ± standard deviation (S.D.) for 3 V and 3.4 mA stimulation amplitudes for all axon diameters simulated. An EF isolevel of 0.2 V/mm corresponding to an axon diameter of approximately 4 µm was selected to compare the activation distances between the leads.

The EF isolevel 0.2 V/mm was superimposed on the preoperative 3T MRI, and visualized at the axial, sagittal and coronal planes. The isocontours for each simulation were extracted in order to measure the maximal distance (mm) from the isocontour to the centre of the active electrode. A program in MatLab was developed for this purpose. COMSOL’s integration function was used to calculate the volumes (mm^3^) inside the 0.2 V/mm EF isosurfaces for all leads. Relative differences in percentages were calculated for voltage and current control in order to compare the results for (I) homogeneous vs. patient-specific models; (II) 3389 lead vs. leads 6148, 6180 and SureStim1.

## 3. Results

### 3.1. Neuron Model Simulations

The selection of an EF isolevel of 0.2 V/mm was supported by the neuron model simulations (Figure 3) for an axonal diameter of 4.0 µm in both homogenous and heterogeneous tissue models (Figure 4a,b).

Figure 5 presents the activation distances at the posterior direction for all four leads in voltage (Figure 5a,c) and current modes (Figure 5b,d), as well as homogeneous (Figure 5a,b) and patient-specific (Figure 5c,d) brain models. Plots of the other three directions (anterior, lateral, medial) are part of the Appendix A (Figure 12, Figure 13 and Figure 14).

### 3.2. Homogenous vs. Patient-Specific Models

The electric field around the 3389 lead was compared for homogeneous and patient-specific models at the ZI and the VIM. Figure 6 shows the influence of the heterogeneity of the tissue. The EF extension for homogeneous tissue model was 3.3, 3.6 and 3.4 mm at the axial, sagittal and coronal planes, respectively, while for the patient-specific model the extension varied from 3.3 to 3.9 mm. The average EF distribution was 12% larger in current mode. This was valid for the three directions explored, in both anatomical regions investigated. The EF volumes achieved at the ZI were larger than those at VIM. The volumetric difference between targets (Table 1) was higher in current mode (12%) than in voltage mode (5%).

### 3.3. Lead Comparison

The EF volumes (Figure 7 and Figure 8) within the 0.2 V/mm isosurface were approximately 49% larger for current controlled stimulation than for voltage mode. The relative difference of the EF volumes between the ZI and the VIM are shown in Table 2, for voltage and current controlled stimulation, respectively.

The electric field simulated for the four different lead designs was visualized at axial, sagittal and coronal planes crossing at the centre of each lead in the middle of the active contacts (Figure 9). The maximum extension of the 0.2 V/mm isocontour in voltage mode was achieved with lead 6148 while for current lead SureStim1 presented the largest EF extension. An example of the maximum EF spatial extension at the ZI, measured from the lead axis, is shown in Table 3.

### 3.4. Patient-Specific Stimulation Amplitude Setting

The patient-specific simulation for the ZI using lead 3389, is presented in Figure 10. The equivalent amplitude for the patient-specific voltage of 1.6 V was 1.3 mA in current mode. This value achieved the most similar EF extension (~2.5 mm) and volume (46 mm^3^) (Figure 10).

### 3.5. Steering Function

The EF volumes within the 0.2 V/mm isosurface and the corresponding isocontours (Figure 11, Table 4) show that the EF distribution was notably different between operating modes for both leads. The spatial extension of the electric field was around 50% smaller in voltage mode. The smaller EF volumes are shown in Figure 11a,b. The axial and coronal views (first and third columns of Figure 11e) show the steering effect on the EF. The large EF distribution achieved by 3.4 mA did not show the steering effect (second and fourth columns of Figure 11e). The diamond configuration used for SureStim1 (1.6 mm^2^ surface area) achieved larger EF volume (Figure 11b) than that using one contact of the 6180 lead (1.8 mm^2^) for voltage mode. The opposite relation was observed in current mode, where 6180 lead achieved a larger EF volume (Figure 11d).

## 4. Discussion

In this study, the influence on the electric field around DBS leads, from surrounding tissue and lead design, has been investigated by means of computer simulations. Both symmetrical and steering functions were considered and compared in current and voltage modes.

### 4.1. FEM and Neuron Modelling

The FEM models in this study have considered constant voltage and current amplitudes instead of the actual biphasic pulse used for the stimulation. This implies a quasi-static solution for the electric potential decoupled from the capacitive, inductive and wave propagation effects. Nevertheless, the conductivity values, for this FEM simulation method, took into consideration the frequency and pulse length components of the stimulation pulse [20]. The comparison of the leads relied on setting as many variables (e.g., isolevel, neuron diameter, pulse width, frequency, tissue variability, time points) to constant values. This results in an evaluation in a fixed environment where the differences in the achieved EF is sufficient to assess the leads. The selection of the 0.2 V/mm isolevel was initially based on previous studies by Hemm et al. [5] and Åström et al. [14]. However, the FEM model used by Åström did not consider the PES and used a homogenous model with a slightly different conductivity value for the grey matter. Therefore, the electric potential lines imported to the neuron model showed minor deviations compared to the previous study. The neuron simulations in the present study indicated that for neurons of 4 µm diameter, a 3 V drive potential reaches an activation distance of 3.2 mm. These results were tested against the FEM simulated EF extensions for one direction and plane, which support the EF isolevel of 0.2 V/mm in the patient-specific model.

Neuron diameter results were in the range of those found in [14,23,24,25], with consideration for driving parameter variations i.e., pulse width. FEM simulated EF extensions ranged from 3.3 to 3.5 mm in voltage mode. The FEM simulation values would imply a neuronal diameter between 4 and 5 µm. These diameters are at present a best guess at the true neuronal diameters in the vicinity of the electrode and should encompass a range of small diameters. As expected, the activation distance for the patient-specific model is distinct from that of the homogeneous model for all leads (Figure 5, Figure 12, Figure 13 and Figure 14).

A variation of 1 mm in activation distance with the working assumption of a 4 µm diameter neuron would result in an increase in neuron recruitment of approximately 250 extra neurons along a radius. For example, if the activation distance increases by 1 mm from 3 mm, the recruitment volume would change to the power of three, i.e., neuron activation expands significantly. An equivalent decrease in activation distance would result in a possible reduction of activated neurons along any radius from the centre of the volume. Calculating the activation distance in different directions (medial, posterior, anterior, lateral) allowed us to assess the influence of the lead’s angle (trajectory) and thus the sensitivity to the direction (Figure 5, Figure 12, Figure 13 and Figure 14).

### 4.2. Homogeneous vs. Patient-Specific Tissue Models

The initial part of the study encompasses a comparison between homogeneous and patient-specific models for the standard 3389 lead in voltage and current modes. Several studies have shown the impact of the anisotropy and heterogeneity of the brain model. The McIntyre group [17] compared the axonal activation during monopolar DBS for different types of models, and concluded that simplistic models, such as the homogeneous model, overestimate the extent of neural activation. Åström et al. [12] observed an alteration of the electric field when the brain was modelled as heterogeneous isotropic tissue as opposed to homogeneous grey matter. These studies, however, were limited to voltage control stimulation. The novelty of the present study relies on the inclusion of current controlled stimulation. Our results show distinct behavior for each operating mode. The 3389 lead EF volume is smaller for the patient-specific model than for the homogeneous model in voltage mode. In current mode, on the contrary, the volume is larger. Furthermore, when comparing the EF volume between targets, the EF difference is larger in current stimulation (12% vs. 5% for voltage). The interest in using current controlled stimulation [26] partly relies on the consideration that it is the capacitive current that determines the neuronal effect; maintaining a constant current presumably would avoid the reprogramming of the DBS which normally occur for voltage controlled systems due to changes in the tissue impedance around the lead [4]. In agreement, the review by Bronstein et al. [24] considers the stimulation field as the electrical delivery which is a function of the voltage divided by the impedance, i.e., current.

The fundamental difference of this study is that the leads are evaluated in terms of the achieved EF and not in the current delivery. The results are numerically obtained considering Equation (1), where the EF is directly proportional to the current density and inversely proportional to the electrical conductivity obeying Ohm’s law. The anisotropy of the tissue has not been included in the model, nevertheless with the introduction of tractography and white matter tracing [7,27], this feature will be important to consider in future simulations. Given that white matter is anisotropic, then the white matter tracing can help make the tissue conductivity classification even better.

### 4.3. DBS Leads Comparison

In the second part of the study, only patient-specific models were used to investigate the EF achieved by four different lead designs operated in voltage and current modes. The results of the simulations showed a very similar EF distribution around each lead, however SureStim1 showed a more spherically shaped EF distribution. In general, the EF extension and volume were higher using current mode and lower for voltage mode. The total current delivered by the electrode is determined by the electrode surface area and the average of the current density. Thus, applying a fixed total current of 3.4 mA to a smaller active area, as SureStim1 lead (3.12 mm^2^) increases the current density, leading to an increase of the EF (Equation (1)). An experimental evaluation of segmented electrodes by Wei and Grill [28] showed that the electrode impedance was inversely proportional to its surface area. This implies that larger contacts would require higher current intensities to achieve the same EF than smaller electrodes. Another example of this behavior is lead 6148, which electrodes have the largest surface area (6.6 mm^2^) achieving the smallest EF in current mode.

Several studies have compared the conventional steering leads either experimentally [29] or based on computer models [2,30,31]. In the experimental study, Contarino et al. [29] temporally inserted a 32 contact lead (similar to SureStim1) which was set with different configurations and current stimulation amplitudes ranging from 0.5 to 8 mA. The steering lead was then replaced by the permanent conventional 3389 lead. The performance of the steering lead was assessed by the current thresholds required to either induce side effects or clinical benefits in comparison to the conventional lead outcome in patients undergoing DBS surgery. By setting 12 consecutive contacts, the Contarino group observed equivalent current thresholds between the steering and the conventional leads. In the present study, eight consecutive electrodes achieved a larger EF volume than the 3389 lead when set to 3.4 mA, implying that choosing 12 contacts instead of eight would increase the difference with the conventional lead even more. This result reflects the influence of the smaller electrodes of SureStim1 lead.

Other computer based studies compared the steering and the conventional leads operated in either voltage or current mode. Martens et al. [2], for instance, investigated a lead of 64 contacts using eight consecutive contacts set to 2.6 mA and observed that a potential field distribution very similar to the generated by the standard ring electrode; our results showed a larger EF for SureStim1 in current mode. The difference between Martens’ model and ours, is the brain model. While they consider homogeneous tissue with a single value of conductivity (0.1 S/m), we include a heterogeneous matrix of electrical conductivities. Dijk et al. [28] also compared the steering lead (SureStim1) to the conventional 3389 lead, however they quantified the stimulation effect in terms of the maximum amount of subthalamic nucleus (STN) cells activated based on axon models. They observed equivalent results between the standard and the directional lead by activating 12 consecutive contacts on the latter lead. In addition, this group used biphasic current pulses and neuron diameters of 5.7 µm. Due to the differences in the evaluation methodology and the model itself, our results are not directly comparable to the results of other groups.

### 4.4. Patient-Specific Stimulation Amplitude Setting

For the actual amplitude programmed, 1.6 V, the EF volume within the 0.2 V/mm isosurface was around 46 mm^3^, and the extension was approximately 2.5 mm measured from the lead axis in all directions. The clinical effect was satisfactory according to the patient journal, however, considering the dimensions of the ZI which has an elongated shape of approximately 2 mm (latero-medial) and 4–5 mm (anterio-posterior), a symmetrical stimulation field could possibly be improved by steering the field in the desired direction. The current amplitude required to achieve the same EF was 1.3 mA, which in comparison to the equivalence for the homogeneous model, indicates a larger impedance for the patient-specific model.

### 4.5. Steering DBS Leads

The steering function of lead 6180 and SureStim1 was evaluated in voltage and current mode. As for the symmetrical configuration, the EF was larger for current control. Setting 3.4 mA to a single contact of lead 6180 (1.8 mm^2^) and to 4 contacts in SureStim1 (1.6 mm^2^) derived in a large EF which did not show the directionality of the configuration. By reducing the current stimulation amplitude to 1.3 mA, it was possible to see the same steered profile as that for 3 V. The reason for this behavior is also due to the increase of the current density for smaller contact surface areas. In a similar way, the directionality of the configuration is not observable by lower EF isolevels. For instance, an isolevel of 0.1 V/mm did not show the steered field of 3 V. This is particularly interesting due to the uncertainty of the EF intensity required to activate neighboring neurons. The EF volumes achieved by each lead in the steering configuration do not follow the rationale of smaller surface area, larger EF due to higher current density. One of the reasons for this behavior could be that the active contacts do not have the same orientation. While the electrodes for SureStim1 are oriented towards the anterior part of the model, the single active contact for lead 6180 is oriented towards the lateral side. In voltage mode, the larger EF volume obtained with smaller surface areas may respond to the increase of the current density due to the higher number of edges [28]. Further investigations focused on different configurations for the steering leads are necessary to satisfactorily assess the performance of directional leads.

## 5. Conclusions

In conclusion, the use of brain models based on patient-specific images and the comparison of two operating modes have enhanced the assessment of the influence from the different lead designs on the EF with a fixed isolevel. The results showed that the EF distribution is influenced by the heterogeneity of the tissue for both operating modes. Computer models can visualize the electric field and thus further increase understanding when switching the stimulation settings, lead designs and inter and intra-patient conductivity variability.

## Figures and Tables

**Figure 1 brainsci-06-00039-f001:**
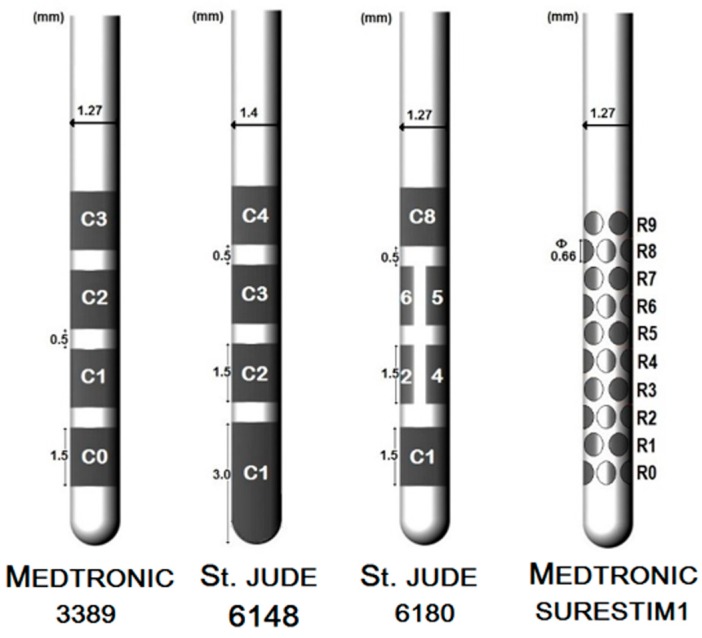
Representation of the conventional and the steering field leads.

**Figure 2 brainsci-06-00039-f002:**
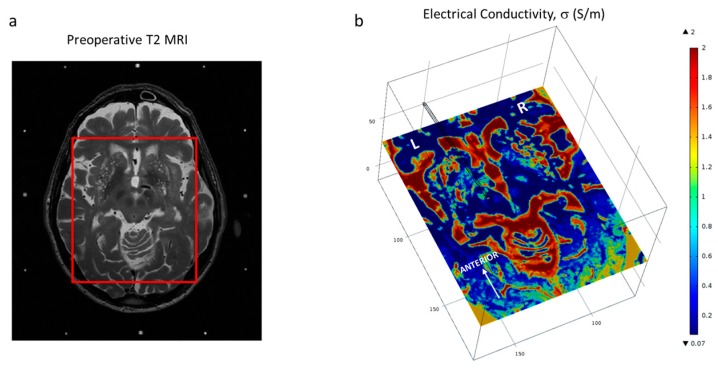
(**a**) Demarcation of the region of interest on the patient T2 MRI dataset (cauda-cranial point of view) and (**b**) Brain model displaying one slice of the interpolated conductivity matrix (cranio-caudal point of view) and the trajectory of the lead. Axial images displayed at the level of the ZI.

**Figure 3 brainsci-06-00039-f003:**
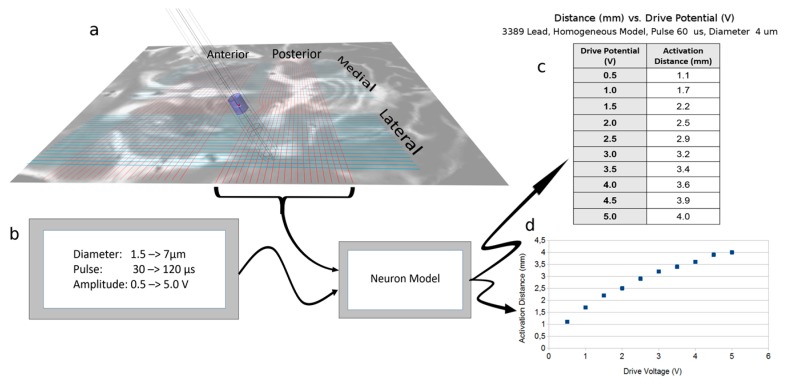
Neuron model application and single calculation run. (**a**) The voltage gradient extraction lines generated from FEM (COMSOL) simulation. The posterior lines have been replaced by the real potential values along the lines, as can be seen by the deviation of the line close to the electrode; (**b**) Input to the neuron model and the model block [14]; (**c**) Data points output from the Neuron model for the 3389 lead, with the specific input parameters of FEM output (homogeneous model and 3389 lead), pulse length of 60 µs, and neuronal diameter of 4 µm. The output is the distance from the surface of the lead to the distance where activation no longer happens; (**d**) The graphical implementation of the one data set.

**Figure 4 brainsci-06-00039-f004:**
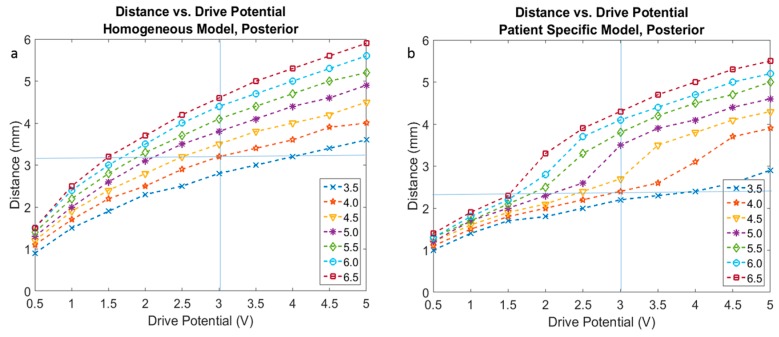
Activation distance plots based on FEM analysis for voltage driven lead 3389 with fixed parameters of 60 µs pulse width, drive potentials range of 0.5 to 5 V, and neuron diameters ranging from 3.5 µm to 6.5 µm. (**a**) Homogeneous tissue model and (**b**) patient-specific tissue model.

**Figure 5 brainsci-06-00039-f005:**
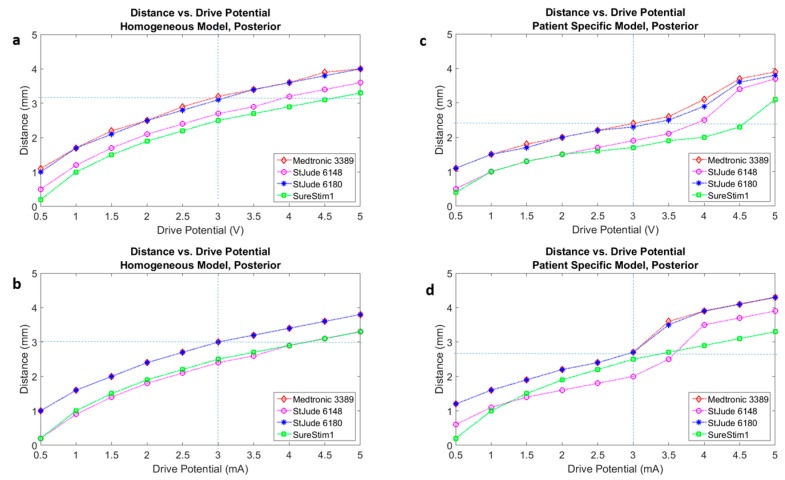
Activation distances for four leads mapped onto a single plot under the same test conditions of 60 µs pulse width, neuron diameter of 4 µm, configuration of all leads in 3389 lead single ring equivalent. (**a**) Homogeneous tissue model with voltage driven electrode; (**b**) Homogeneous tissue model with current driven electrode; (**c**) Patient-specific tissue model with voltage driven electrode; (**d**) Patient-specific tissue model with current driven electrode.

**Figure 6 brainsci-06-00039-f006:**
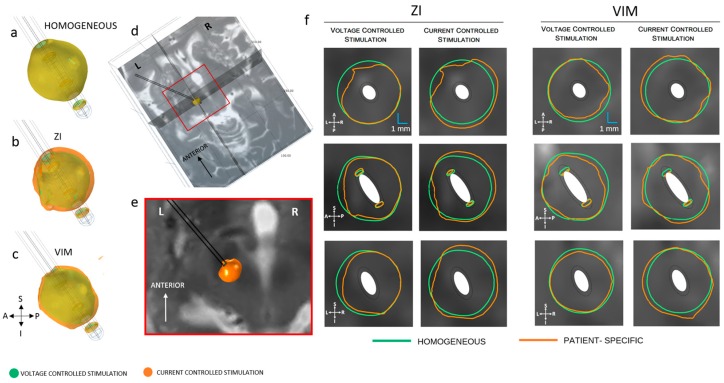
Electric field (EF) distribution (0.2 V/mm) in voltage and current control stimulation mode. (**a**) Homogeneous model (**b**) patient-specific model, ZI and (**c**) VIM; (**d**) Axial, sagittal and coronal cut planes, crossing at the middle point of the active contact (**e**) closer view of the axial plane of the preoperative MRI at the ZI and (**f**) electric field isocontours (0.2 V/mm) of lead 3389 for homogeneous and patient-specific brain models. EF obtained at 3 V (first and third column) and 3.4 mA (second and fourth column). A: anterior, P: posterior, S: superior, I: inferior, L: left, R: right.

**Figure 7 brainsci-06-00039-f007:**
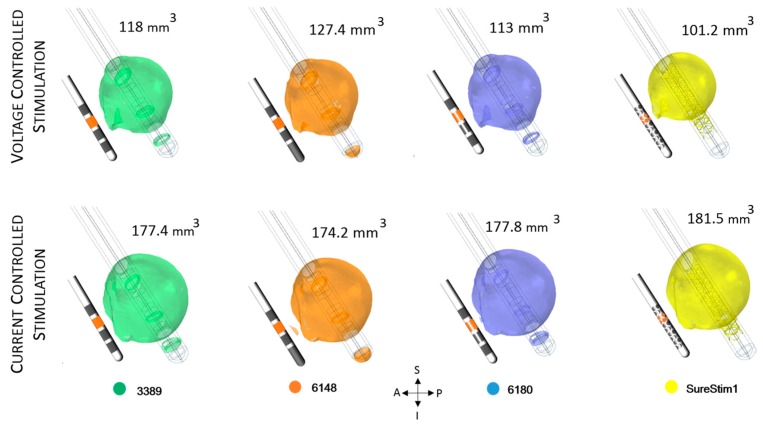
Electric field (EF) simulated at ZI for each lead depicted with an isosurface of 0.2 V/mm. Active contacts (shown in orange in each lead schematic) set to 3 V (first row) and 3.4 mA (bottom row). EF volume within the selected isosurface shown to the right of the lead. A: anterior, P: posterior, S: superior, I: inferior, L: left, R: right.

**Figure 8 brainsci-06-00039-f008:**
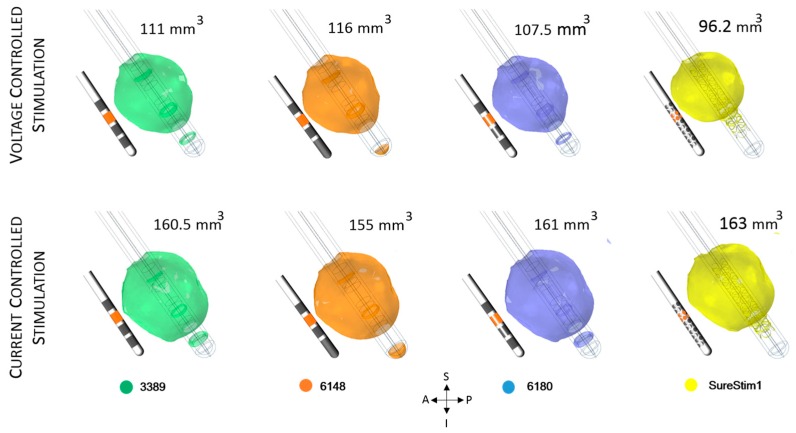
Electric field (EF) simulated at VIM for each lead depicted with an isosurface of 0.2 V/mm. Active contacts (shown in orange in each lead schematic) set to 3 V (first row) and 3.4 mA (bottom row). EF volume within the selected isosurface shown to the right of the lead. A: anterior, P: posterior, S: superior, I: inferior, L: left, R: right.

**Figure 9 brainsci-06-00039-f009:**
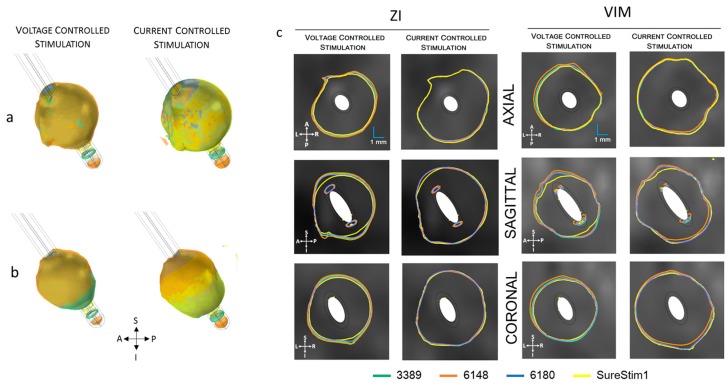
Electric field (EF) 0.2 V/mm isosurfaces achieved by each lead superimposed for each EF distribution of each lead operated in voltage (3 V) and current (3.4 mA). (**a**) EF isosurfaces at ZI in voltage (left) and current (right); (**b**) isosurfaces at VIM for voltage (left) and current (right); (**c**) Isocontours (0.2 V/mm) at the axial, sagittal and coronal planes. The cut planes for visualization were placed at the coordinates of the middle point of the active contacts. A: anterior, P: posterior, S: superior, I: inferior, L: left, R: right.

**Figure 10 brainsci-06-00039-f010:**
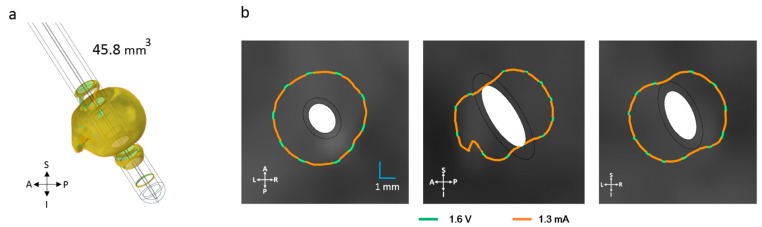
(**a**) Electric field (EF) distribution when the contact is set to 1.6 V and the equivalent current 1.3 mA (superimposed); (**b**) Isocontours for voltage and current superimposed. The maximal EF extent using an isolevel of 0.2 V/mm measured from the middle point of the active contact was 2.5 mm in all planes for both operating modes. A: anterior, P: posterior, S: superior, I: inferior, L: left, R: right.

**Figure 11 brainsci-06-00039-f011:**
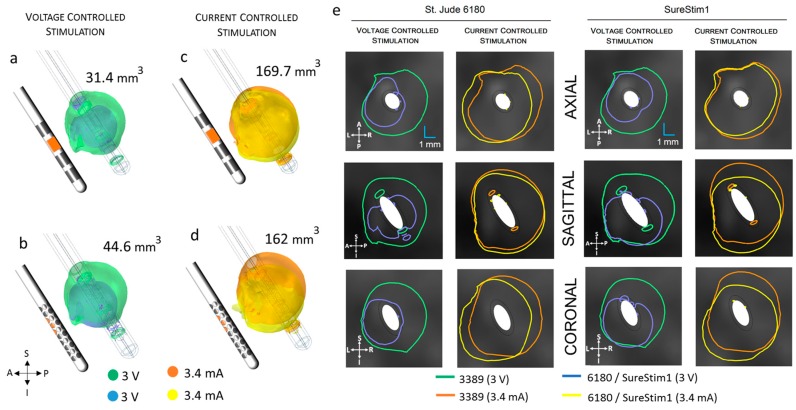
Comparison of the electric field (EF) isosurfaces (0.2 V/mm) at the zona incerta between the standard lead 3389 and the steering leads (active contacts shown in orange in the lead schematic). EF superimposed for lead 3389 (green/orange volumes) and (**a**) lead 6180 contact 5 active; (**b**) SureStim1 lead using the diamond configuration, operated in voltage mode (smaller volumes in blue); (**c**) Lead 6180 and (**d**) SureStim1 setting the contacts to current mode (EF volumes in yellow); (**e**) EF isocontours (0.2 V/mm) at the axial, sagittal and coronal planes for both leads operated in voltage and current mode. A: anterior, P: posterior, S: superior, I: inferior, L: left, R: right.

**Table 1 brainsci-06-00039-t001:** Homogeneous and patient-specific electric field (EF) volumes (<0.2 V/mm isosurface) achieved with different operating modes and the relative difference between each target and the homogeneous volumes.

Model	Voltage	Current	Voltage	Current
	Volumes (mm^3^)	Volumes (mm^3^)	Difference (%)	Difference (%)
HOMOGENEOUS	144	144	0	0
ZI	118.0	177.4	−18.0	22.9
VIM	111.0	160.5	−23.2	11.4

**Table 2 brainsci-06-00039-t002:** Electric field (EF) volume determined by the 0.2 V/mm isosurface achieved by 3 V and 3.4 mA. Relative difference between the targets calculated for each operating mode.

Lead	ZI (mm^3^)		VIM (mm^3^)		Relative Difference (%)	
	Voltage	Current	Voltage	Current	Voltage	Current
3389	118.0	177.4	111.0	160.5	5.9	9.5
6148	127.4	174.2	116.0	155.0	8.9	11.0
6180	113.0	177.8	107.5	161.0	4.9	9.4
SureStim1	101.2	181.5	96.2	163.0	4.9	10.2

**Table 3 brainsci-06-00039-t003:** Maximum spatial extension (mm) of the 0.2 V/mm electric field isolevel achieved at each plane for voltage (3 V) and current (3.4 mA) controlled stimulation for all leads. Measurements performed at the ZI.

Plane	3389		6148		6180		SureStim1	
	Voltage	Current	Voltage	Current	Voltage	Current	Voltage	Current
AXIAL	3.34	3.85	3.46	3.84	3.29	3.86	3.23	3.94
SAGITTAL	3.40	3.87	3.50	3.85	3.35	3.90	3.17	3.90
CORONAL	3.50	3.83	3.55	3.80	3.32	3.84	3.23	3.88

**Table 4 brainsci-06-00039-t004:** Maximum spatial extension of the 0.2 V/mm electric field isolevel (mm) achieved by steering configurations. Relative difference between operating modes calculated for each lead.

Plane	6180		SureStim1		Relative Difference (%)	
	Voltage	Current	Voltage	Current	6180	SureStim1
AXIAL	2.80	4.18	2.51	3.65	49	45
SAGITTAL	2.92	3.95	3.18	4.46	36	40
CORONAL	2.68	4.54	3.15	4.69	69	49

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
