# Peer review of "Investigation into Deep Brain Stimulation Lead Designs: A Patient-Specific Simulation Study"

_brainsci, 2016, doi:10.3390/brainsci6030039_

Round 1

Reviewer 1 Report

Dear editor,

The authors have submitted a rigorous and extensive work on the evaluation of multiple DBS leads, for homogeneous and realistic surroundings, for two treatment targets, for two excitation modes. I certainly recommend publication and have actually not been able to pinpoint any shortcoming.

Author Response

 We thank your revision of our manuscript and the comments about it.

Reviewer 2 Report

The authors proposed an in silico patient-specific study to compare the electric field (EF) distributions of 4 deep brain stimulation (DBS) lead configurations, operated using both current and voltage modes, at equivalent amplitudes. Two leads investigated were conventional, cylindrical contacts creating symmetric stimulation fields while the other two were capable of steering the stimulation fields.  Simulations were performed using the finite element method (FEM) and the influence of both homogenous and heterogenous brain tissue properties at the target sites (zona incerta and ventral intermediate nucleus) were investigated. To facilitate effective comparisons between simulations an electric field isolevel was chosen based on neuron model simulations, conducted for each lead configuration. EF simulations were shown to be sensitive to the heterogeneity of brain tissue and the operating mode employed, while equivalent lead configurations resulted in similar EF distributions. The authors show that the steering lead configurations resulted in large EF volumes when operated in current mode and conclude that the differences between lead designs are enhanced when heterogenous tissue properties and current-controlled stimulation are employed.

Broad Comments:

The paper is well written and easy to follow. The authors have conducted a comprehensive set of experiments which are described in detail. The conclusions are consistent with the results presented.

(1) Section 2.2 does not include any information on the mesh generation algorithm used for all simulations. There is no information regarding the overall mesh size/density used in the simulations. The authors mention the use of the multi-physics solver available in COMSOL but do not discuss the reason for this choice and do not provide any details regarding the solver itself. Additionally, there is no discussion of a mesh convergence study which is important to ensure that the final simulation results are mesh independent.

(2) Throughout the paper the authors refer to heterogenous brain tissue properties as patient-specific tissue models. It is unclear how the authors estimate/employ personalised brain tissue properties as the electrical conductivity values used for white matter, grey matter and CSF are taken from literature. The authors should clarify whether the FE model geometry (i.e. distribution of WM, GM and CSF regions) and lead configurations are personalised or whether the tissue properties themselves are patient-specific.

(3) Segmentation of the pre-operative brain MRI was performed by intensity thresholding to distinguish between WM, GM and CSF. However, thresholding techniques are well known to be inadequate in the presence of intensity inhomogeneities, common to brain MRIs and at capturing partial volume effects at tissue boundaries. There is no description of how these issues were addressed and their potential implications on the electrical conductivity values used in such regions. As a related comment, it is unclear why the authors do not employ open source software tools such as FSL-FAST (http://fsl.fmrib.ox.ac.uk/fsl/fslwiki/FAST) for example, to generate tissue maps/segmentations from the pre-operative MRI.   

Minor Comments:

(1) The abbreviation VIM is used in the abstract before definition and should be included either in the abstract or in the keywords, similar to ZI.

(2) In section 2.2.2 line number 136, the authors mention a minimum tetrahedral element quality of 0.136 but do not highlight which quality measure this refers to. 

(3) There is no description of Fig. 11(e) in the caption and no reference to it in the main text.

Reviewer 3 Report

Thank you for the opportunity to review "Investigation into Deep Brain Stimulation Lead Designs: a Patient-specific Simulation Study". This manuscript describes a computational study to assess the efficacy of different DBS electrode designs. Given the current interest in DBS and the evolving hardware in this area, this could be of great interest to the readers. However, in its current form I feel that the manuscript needs revision.

Specific comments:

My main concern is about the description of the methodology used in this study. Specifically, on page 3 the authors refer to an "in-house" program ELMA, which they reference but do not describe at all. It is essential to explain what this program does exactly even if concisely. Even for someone with experience of this field, it is very unclear how the model was actually constructed. The authors need to revise this section and make the link from imaging data to 3-dimensional model with heterogenous conductivity values much more transparent. Figure 2b hints that the imaging data was restricted to a rectangular region (on each slice) and the grey scale values were used as conductivity values, but this is not clearly explained. Another figure of the modelling process is needed, such as an image of the 3-dimensional brain model.

A related comment about the description of the FEM modelling and simulation section of the Materials and Methods. The authors describe a number of lead designs and later a number of stimulation settings. Although they do provide a figure with the designs, I feel that a table describing the differences and simulation settings might be useful.

On page 3 the authors say that the first contact was placed at the lower point noted by the surgeon. It is unclear why the tip of the electrode was not placed here.

It is also unclear what the difference between the two patient-specific simulation types (ZI and VIM) is? Did they move the electrode in the anatomic model for each target.

For the description of the electric field equation solved in the model, it would be good to see boundary conditions set out mathematically to avoid any ambiguity in the text.

Figure 3 is not clear to me at all. I am not sure what the different coloured lines represent, and it might be helpful to orientate this with respect to the anatomy.

Was the selection of the simulated stimulation parameters arbitrary, or is it based on the patient's specific outcome? I.e. did the patient have a good clinical outcome on any of these settings? Could this be used to validate any of the modelling results?

Am I right that the comparison of voltage and current controlled stimulation is based on choosing a current amplitude that results in a similar EF to the voltage controlled stimulation (as described on page 9)? 

Round 2

Reviewer 3 Report

I am satisfied with the authors responses to my comments, but please ensure that where further explanation/clarification has been provided, i.e. comments 3,4,7 and 8, these have all been incorporated into the final article for the benefit of the journal's readers.

Author Response

Thank you for your suggestions, the manuscript has been updated accordingly. The text added has been marked in yellow.

Below you will find the answer to the comments you pointed out and the description of the changes including the number of the lines modified (line numbers are considered when “simple markup” is selected within the Track Changes tool menu in Microsoft Word).

Comment 3: The procedure to obtain the coordinates of the contacts is described in lines 77-80. The artefact coordinates obtained from the CT scan correspond to the first contact which was used as the reference to place the first electrode of lead 3389.

Comment 4: The statement moving the leads accordingly, approximately 4 mm along the trajectory)" is now included in the paper (line 176)

Comment 7: The amplitude chosen (3 V) is within the range (1- 4 V) of the typical values clinically used and also used in a previous study by us, please also check reference 11 (Alonso 2015). Regarding the amplitude of 1.6 V, the clarification below has been added (lines 179-180)

with the actual stimulation 1.6 V, set four and a half weeks after implantation, which relieved the patient’s symptoms.

Comment 8: Already clarified in section 2.2.2 with a reference to our previous study where the methodology is described (lines 138-141).